# A Novel Identified Necroptosis-Related Risk Signature for Prognosis Prediction and Immune Infiltration Indication in Acute Myeloid Leukemia Patients

**DOI:** 10.3390/genes13101837

**Published:** 2022-10-11

**Authors:** Yong Sun, Ruiheng Wang, Shufeng Xie, Yuanli Wang, Han Liu

**Affiliations:** 1National Research Center for Translational Medicine at Shanghai, Ruijin Hospital, State Key Laboratory of Medical Genomics, Shanghai Institute of Hematology, Shanghai Jiao Tong University School of Medicine, Shanghai 200025, China; 2Fujian Provincial Key Laboratory on Hematology, Fujian Medical University, Fuzhou 350122, China

**Keywords:** acute myeloid leukemia, multivariate Cox regression, necroptosis, immune infiltration, prognostic model

## Abstract

AML ranks second in the most common types of leukemia diagnosed in both adults and children. Necroptosis is a programmed inflammatory cell death form reported to be an innate immune effector against microbial and viral pathogens and recently has been found to play an eventful role in the oncogenesis, progression, and metastasis of cancer. This study is designed to explore the potential value of necroptosis in predicting prognostic and optimizing the current therapeutic strategies for AML patients. We collected transcriptome and clinical data from the Cancer Genome Atlas (TCGA) and the Genotype-Tissue Expression (GTEx) databases and selected necroptosis-related genes with both differential significance and prognostic value. Six genes (*YBX3*, *ZBP1*, *CDC37*, *ALK*, *BRAF*, and *BNIP3*) were incorporated to generate a risk model with the implementation of multivariate Cox regression. The signature was proven to be an independent prognostic predictor in both training and validation cohorts with hazard ratios (HRs) of 1.51 (95% CI: 1.33–1.72) and 1.57 (95% CI: 1.16–2.12), respectively. Moreover, receiver operating characteristic (ROC) curve was utilized to quantify the predictive performance of the signature and satisfying results were shown with the area under the curve (AUC) up to 0.801 (3-year) and 0.619 (3-year), respectively. In addition, the subtyping of AML patients based on the risk signature demonstrated a significant correlation with the immune cell infiltration and response to immunotherapy. Finally, we incorporated risk signature with the classical clinical features to establish a nomogram which may contribute to the improvement of clinical management. To conclude, this study identified a necroptosis-related signature as a novel biomarker to improve the risk stratification, to inform the immunotherapy efficacy, and to indicate the therapeutic option of targeted therapy.

## 1. Introduction

AML is a highly heterogeneous malignant tumor featuring aberrant proliferation and accumulation of immature myeloid hematopoietic stem or progenitor cells in bone marrow, leading to the impairment of normal hematopoietic function [1]. The proportion of AML in all types of leukemia almost reaches 70% and 30% in adult patients and pediatric patients, respectively. The pathogenesis of AML has not been fully clarified and so far, it has been generally believed that potential mechanisms include gene fusions, dysregulated signal pathways, altered bone marrow microenvironment, etc. [2,3,4,5]. While recognized as the standard therapy for the AML, chemotherapy remained a suboptimal therapeutic option as more than two-thirds of adult AML patients would suffer relapse after the primary remission [6]. Therefore, novel and synergistic therapeutic approaches are urgently needed to improve the clinical outcomes.

In recent years, the development of immunotherapy has gained wide attention and holds great promise for AML treatment. As one of the immunotherapy regimens, antibody therapy exerts anti-tumor effects based on the according ability of the antibody, such as unconjugated antibodies prompting NK cells antibody-dependent cell-mediated cytotoxicity (ADCC), or bi- and tri-specific antibodies engaging NK cells or T-cells to efficiently improve the cytotoxicity against target AML cells, etc. [7]. While clinical outcomes of several unmodified mAbs exhibited limited benefits, studies of modified antibodies such as BI836858 are under way [8]. Instead of targeting specific leukemia cells, vaccines and checkpoint blockade strategies are centered on the reactivation of antileukemia immunity. Both peptide- and DC-based vaccines have primarily documented antileukemia immunological responses and improved survival outcomes in different clinical trials [9,10]. Several studies exploring novel vaccine strategies including optimizing immunostimulatory properties of DCs are still in progress [11,12]. Immune checkpoint blockade (ICB) has achieved durable responses in the patients with melanoma and other solid tumors. While most studies concerning CPIs in AML are still in the early stage of clinical trials, the combination of *PD-1* or *CTLA4* blocking antibody with HMAs has already produced encouraging responses [13]. In the future, it can be safely concluded that combination regimens of immunotherapy with chemotherapeutics or other carcinogenic pathway inhibitors will outscore monotherapy as a prime option.

Characterized by the rapid membrane permeabilization and properties of inflammation, necroptosis is defined as a combination of apoptosis and necrosis due to the mechanistic and morphological resemblance [14]. The occurrence of necroptosis was initially thought to be RIPK1-dependent (receptor-interacting protein [*RIP*] kinase 1) triggered by a plethora of upstream stimuli including tumor necrosis factor receptor (*TNFR*), *Fas*, etc. Recently it has been discovered that in certain cases, Toll like receptor 3/4 (*TLR-3/4*) and Z-DNA binding protein 1 (*ZBP1*) can directly induce cells to undergo necroptosis without the involvement of *RIPK1*, which identifies *RIPK3* and its substrate mixed-lineage kinase-like (*MLKL*) as more specific molecular markers of necroptosis [15,16]. Upon the phosphorylation of *RIPK3* through the interaction with *RIPK1*, *MLKL* is phosphorylated and oligomerized and then translocated to the cell membrane to form pores. The influx of particular ions leads to the cell swelling, membrane lysis, and eventually cell demise [17]. Necroptosis was primarily deemed as an indispensable role in the innate immunity to fight against viral and bacterial infections, while a growing number of studies indicated its double-edged sword role in a lot of tumors. Liu et al. [18] found that the necroptosis inhibitor NSA (necrosulfonamide) has a remarkable suppression effect on tumor progression in a xenograft model, suggesting the potent pro-tumor function of necroptosis. In contrast, low or undetectable expression of *RIPK3* is witnessed in numerous cancer cell lines. Moreover, decreased expression of *RIPK3* is related with a shortened OS in patients with breast cancer [19]. The unique inflammatory microenvironment formed by the cytokines released accompanied with necroptosis may allow its dual effect in tumors. However, the specific role of necroptosis in AML has not been studied yet.

In this study, we systematically explored the clinical value of genes related to necroptosis in AML as well as potential mechanisms. We compared the expression level of necroptosis-related genes and constructed a risk signature based on the screened genes. We further assessed the prognostic value of the signature and its association with the immune microenvironment. Given the existing results, we consider the signature a reliable prognostic predictor to improve clinical management for AML patients, a potential target for tumor targeted therapy and an adjunct for immunotherapy to improve patient selection.

## 2. Materials and Methods

### 2.1. Data Acquisition and Preprocessing

Normalized mRNA expression data (FPKM value) along with the corresponding clinical features of 151 AML patients were obtained from the University of California Santa Cruz (UCSC) Xena database (https://xenabrowser.net/datapages/) (accessed on 6 April 2022) as a training set. Another dataset from UCSC Xena (RSEM count) integrating the TCGA-LAML and GTEx-whole blood cohorts was acquired to identify the differentially expressed genes (DEGs), which comprised 173 AML samples and 337 normal samples. The microarray data and overall survival (OS) information of the validation set GSE37642 (GPL570) were further derived from the Gene Expression Omnibus (GEO) database (https://www.ncbi.nlm.nih.gov/geo/) (accessed on 6 April 2022). Transcriptome data of GSE6891, GSE114868, GSE111678, and GSE71014 were also collected to verify the stratification ability of risk signature. Eligible samples were screened according to the following three principles:

1. Complete gene expression data without any NA or missing value. 2. Complete survival information and clinicopathological features including patient gender, age, white blood cell (WBC) count, blast cells percentage in bone marrow (BM), and cytogenetic risk. 3. The overall survival time of the patient should be longer than 30 days. After the data processing and removal of the inappropriate samples, 126 samples from TCGA cohort and 124 samples from GSE37642 cohort were incorporated in this study for subsequent analysis.

### 2.2. Clinical Human Samples

A total of 7 peripheral blood (PB) specimens from 3 healthy donors and 4 AML patients were collected at Ruijin Hospital affiliated to Shanghai Jiao Tong University, School of Medicine between October 2021 and April 2022. Collection and use of samples for this study was approved by the Institutional Review Boards from Ruijin Hospital.

### 2.3. Collection of Necroptosis-Related Genes

A total of 111 necroptosis-related genes (NRGs) originating from the MSigDB (http://www.gsea-msigdb.org/gsea/index.jsp) (accessed on 6 April 2022) and formerly reported literature were enrolled in our study. The specific genes are listed in the Appendix A.

### 2.4. Identification of Differentially Expressed Genes

The voom function within the R package “limma” was utilized to identify the differentially expressed genes between the AML and normal samples. The threshold for DEGs was set as adjusted *p*-value < 0.01. The protein–protein interaction (PPI) network analysis was then performed on the generated DEGs in the STRING database (http://string-db.org) (accessed on 8 April 2022).

### 2.5. Establishment and Assessment of a Necroptosis-Related Prognostic Signature

Univariate Cox regression was adopted in order to evaluate the prognostic value of all genes in the training set. Prognostic-related genes (PRGs) were identified with the threshold set as *p*-value < 0.05. Combining PRGs with the previously generated DEGs and NRGs, we defined the overlapping genes that appear in all three sets as prognostic necroptosis DEGs. Multivariate Cox regression was further applied to establish a necroptosis-related prognostic model. The method selected for the stepwise regression for variables screening was both backward and forward. Risk score for each sample was calculated based on the formula below:$$\mathrm{Risk}\;\mathrm{score}=\sum_{i=1}^{n}\beta_{i}e_{i}$$
where $\beta_{i}$ represented the coefficient of $gene_{i}$ from regression results and $e_{i}$ the expression level of $gene_{i}$.

Considering the median risk score a cut-off point, patients were divided into high or low-risk groups and the difference in OS between two subgroups was then assessed via Kaplan–Meier curve. Precision of the established prognostic model was further verified by drawing the ROC curve. All the steps above were implemented using the “survminer”, “survival”, “survivalROC” and other R packages.

### 2.6. Functional Enrichment Analysis

Identical to the process above, DEGs were filtered between the high and low-risk groups with the cut-off|log2FC| ≫ 0.4 and *p*-value < 0.05. Based on the DEGs, GO, KEGG, and GSEA analyses were then conducted to explore the enriched biological signaling pathways in different subtypes using the “clusterProfiler” package. *p*-value < 0.05 was set to indicate an enrichment significance and the results were visualized by the “ggplot2” package.

### 2.7. Characteristics of Immune Cell Infiltration

A total of 40 immune ingredients consisting of immune cells, relative immune factors, and pathways were acquired from the public resource [20]. Single-sample gene set enrichment analysis (ssGSEA) was performed to produce immune scores individually which indicated the degree of immune infiltration using the “gsva” package. CIBERSORT is a linear support vector regression-based algorithm that calculates the content of 22 immune cell subgroup by deconvolution. By carrying out the CIBERSORT, we quantified and compared the different proportions of immune cells between the two subgroups through the Wilcoxon rank-sum test.

### 2.8. RNA Extraction and qRT-PCR

Total RNA was extracted from 7 PB specimens with FastPure^®^ Cell/Tissue Total RNA Isolation Kit (Vazyme Biotech Co., Ltd., Nanjing, China) and then reverse-transcribed to cDNA with HiScript II Q RT SuperMix (+gDNA wiper) (Vazyme Biotech Co., Ltd., Nanjing, China) following the manufacturer’s instructions. The generated cDNAs were amplified in the subsequent qRT-PCR using Eastep^®^ RT Master Mix (Promega, Beijing, China). The gene expression levels were calculated by the 2ΔΔCT method and human GAPDH was used as a housekeeping gene to normalize target genes. Primers used in real-time PCR are available in the Appendix A.

### 2.9. Statistical Analysis

All the data were processed using the R software (version 4.0.3) and related R Bioconductor packages. Log-rank test was used for the statistical comparison of Kaplan–Meier survival curves. The Pearson chi-square test was used to verify the independence of numeration variables. The Shapiro–Wilk test was first used to judge the normality of the measurement data and then either Spearman or Pearson correlation test was used to compute the coefficients accordingly. All *p*-values involved in the statistical tests were double tailed, and *p* < 0.05 was considered to be statistically significant.

## 3. Results

### 3.1. Construction of a 7-Genes-Based Necroptosis-Related Risk Signature

The overall workflow of our study is displayed in the Figure 1. To gain a better understanding of the association of the 111 necroptosis-related genes collected in this study, we performed the PPI network analysis with a 0.9 level (highest confidence) set for the minimum required interaction score (Figure 2A). A total of 27,966 DEGs were identified between AML and normal samples by the voom method within the “limma” package with the screening criteria of adjusted *p*-value < 0.05 (Appendix A). Univariate Cox regression was applied to distinguish genes of significant prognostic value and 5426 PRGs were identified as a result (Appendix A). By merging DEGs with PRGs and necroptosis-related genes, 22 genes were retained and demonstrated via the Venn plot (Figure 2B). The 22 genes were further incorporated into the multivariate Cox regression to construct the prognostic signature with both forward and backward direction selected for the stepwise regression. A total of 6 genes were finally involved and shown by forest plot using the “survminer” package (Figure 2C). The equation for the model generated was as follows:Risk score = *YBX3* × − 0.3676 + *ZBP1* × 0.5756 + *BRAF* × − 0.6369 + *ALK* × 1.7282 + *BNIP3* × − 0.4797 + *CDC37* × 0.5021

Among the 6 genes, *CDC37*, *ZBP1*, and *ALK* were associated with decreased survival time with HRs > 1 (*CDC37*: 1.65, 95% CI: 0.9–3.04; *ZBP1*: 1.78, 95% CI: 1.12–2.83; *ALK*: 5.63, 95% CI: 1.47–21.56) while the remaining 3 genes were considered protective factors with HRs < 1. Real-Time PCR was used to verify the mRNA levels of 6 genes in 3 healthy samples and 4 AML samples in the in-vitro experiment (Figure 2D). Results of all 6 genes were in accordance with the differential expression analysis.

### 3.2. Evaluation of Model’s Prognostic Prediction Ability and Validation of Its Robustness

Samples in the training set were divided into high and low-risk groups according to the median risk score. Kaplan–Meier curve was utilized to compare the survival conditions between the two subgroups and the results witnessed a survival advantage in the low-risk group (Figure 3A, *p* < 0.0001, Log-rank test). To assess the discrimination accuracy of the model, the ROC curve was drawn and AUC for 1-year, 3-year, and 5-year overall survival fluctuated around 0.8 (0.759, 0.801, 0.759, Figure 3C). GSE37642 from the GEO database was further adopted to verify the universal robustness of the necroptosis-related risk signature. The incidence of death was significantly lower in the low-risk group in the validation set (Figure 3B), which was in good agreement with the results in the training set. The AUCs calculated for the validation set of 1-year, 3-year, and 5-year OS were 0.661, 0.619, and 0.620, respectively.

### 3.3. Correlation of Clinicopathological Features with Risk Score

Distribution of risk score in both training and validation sets were exhibited in Figure 4A,B. The overall survival landscape in both sets were displayed in Figure 4C,D and similar conclusion can be reached that risk score was directly proportional to the number of deaths and inversely proportional to the survival time. Principal component analysis (PCA) was performed to evaluate the clustering performance of risk model (Figure 4E,F) by “ggplot2” package and the plot revealed that it could serve as an acceptable separator between the two subgroups. Then, we applied risk model to the training set stratified by different clinicopathological features to confirm the prediction nature of model regardless of the clinical features impacts (Appendix A). The results observed a significant survival advantage in the low-risk group in most subgroups. Extracting and comparing the expression levels of 6 genes between the high and low-risk groups, we found that all 6 genes were differentially expressed (Figure 4G, Wilcoxon rank-sum test), among which *BNIP3*, *BRAF*, and *YBX3* were upregulated in the low-risk group, while *ALK*, *CDC37*, and *ZBP1* were downregulated in the training set, which was consistent with the results obtained in the validation set (Figure 4H). The 6-genes model combined with critical clinicopathological factors including gender, age, WBC count, blast cells percentage in BM, and cytogenetic risk as well as risk score were further presented in a heatmap (Figure 5A) using the “pheatmap” package. Correlations of the clinicopathological factors with risk score were subsequently explored and the results showed that both age and cytogenetic risk had a significant positive correlation with the risk score. An alluvial diagram was used to demonstrate the interrelations within the clinical characteristics in a dynamic way by the “ggalluvial” package (Figure 5B).

### 3.4. Analysis of Specific Functional Pathways Involved in Different Necroptosis-Related Subgroups

Conforming to the method mentioned above, 123 DEGs were identified between the high and low-risk subgroups by the threshold of |log2FC| ≫ 0.4 and *p*-value < 0.05, among which 710 genes were upregulated in the high-risk group while 450 genes were downregulated (Appendix A). To gain a deep insight into the internal regulation network based on the DEGs and to investigate the difference in the enriched pathways involved in functional genes, we conducted GO, KEGG, and GSEA analysis on the upregulated genes in the high-risk group. The KEGG plot (Figure 6B) reflected that significantly enriched pathways in the high-risk group were mainly related with cytokine signaling pathways downstream the viral infections. The cytokines provided by the necroptotic cells could incur a high activation level of innate immunity in the high-risk group, which was confirmed in the GO analysis (Figure 7D) as several immune-related biological processes were identified. Moreover, the classical pathway involved in the leukemogenesis-like myeloid leukocyte proliferation was also enriched and visualized through the GSEA plot along with two selected immune pathways (Figure 6C,E,F). The complete results of GSEA analysis are available in Appendix A.

### 3.5. Exploration of Immune Infiltration Characteristics in High and Low-Risk Groups

Immune cell infiltration in the immune microenvironment was widely considered to play an essential role in tumorigenesis as well as tumor development and had a profound impact on the patients’ clinical outcomes. Here, in order to extensively uncover the immune features in the high and low-risk groups, we employed the ssGSEA analysis based on the 40 immune components containing immune cells, related pathways, and factors gathered from the public literature and ImmPort database. The different immune scores of two subgroups were computed and displayed in a heatmap (Figure 7A,B). Furthermore, we utilized the CIBERSORT algorithm to quantify and compare the proportion of 22 classic immune cells in the high and low-risk groups. As the results in the training set show (Figure 7C), proportions of eosinophils and mast cells resting were significantly higher in the low-risk group, while proportions of monocytes, T cells CD4 memory activated, T cells CD8, and Macrophages M2 were significantly lower. The same results were confirmed in the validation set (Figure 7D) that mast cells resting were upregulated in the low-risk group while monocytes were downregulated. In addition to the results in common, T cells CD8 exhibited a reverse outcome in the validation set and T cells γ delta and Macrophages M0 were downregulated in the low-risk group.

### 3.6. Prediction of Response to Immunotherapy Targeting Immune Checkpoints

The immune cell infiltration results gave us a hint that the risk model-based subgroups might be potential targets for immune therapy. Tumor mutation burden (TMB) refers to the number of somatic nonsynonymous mutations in a specific genomic region and functions as a novel biomarker suggesting the degree of tumor to produce new antigens and predicting the efficacy of ICB therapy. We utilized the “maftools” package to process the somatic mutation data of the patients in the TCGA cohort downloaded from the UCSC Xena and TMB of each patient was subsequently computed. We firstly compared the TMB between the high and low-risk groups and the result showed no statistical difference. Then, we further analyzed the correlation of the six genes in the risk signature with TMB (Figure 8A–F) and the results reflected that the expression level of *BRAF* was significantly positively correlated with TMB, which suggested that patients with high expression of *BRAF* were more likely to response to ICB. Then, we achieved the expression profiles of eight of the most studied checkpoint genes in the TCGA cohort and made the comparisons between the high and low-risk groups in terms of the expression level. The results (Figure 8G) showed that all eight genes except *IDO* were significantly upregulated in the high-risk group in contrast to the low-risk group, which suggested a potent enhanced response to ICB in the high-risk group. To validate the differential expression results of immune checkpoint genes, we repeated the risk stratification and compared expression levels of *PD1*/*PDL1*, *TIM3*, *CTLA4*, as well as *CDC37* in four external GEO datasets (Appendix A). The same results proved the stratification ability of the signature was robust. Tumor immune dysfunction and exclusion (TIDE) was a computational framework quantifying the immune evasion ability of tumors by modeling two major mechanisms of the tumor immune evasion: the induction of T cell dysfunction in tumors and the prevention of T cell infiltration in tumors. A high TIDE score represented a relative strong immune evasion ability of tumor, which in turn suggested a poor outcome of immunotherapy, vice versa. In this study, we adopted the TIDE algorithm to generate a TIDE score individually in the TGCA cohort and correlation between the necroptosis-related risk score and TIDE score was further investigated. The results depicted in the Figure 8H showed a significant negative correlation between the risk score and TIDE score, which indicated that patients with higher risk score were more likely to benefit from the immunotherapy. Moreover, patients were classified into responder and non-responder to immunotherapy two groups and the risk score between the two subgroups showed a significant difference, which supported the predictive role of risk score in immunotherapy.

### 3.7. Application of the Necroptosis-Related Signature to Building an OS Prediction Nomogram for Clinical Use

Univariate and multivariate Cox regression analysis were performed on the patients in the TCGA cohort in an attempt to assess the validity of the necroptosis-related risk score incorporated with several other primary clinical factors in predicting prognostic. The univariate analysis indicated that age, cytogenetic risk, and risk score were significant unfavorable factors associated with OS (Figure 9A). Then, the above three factors were continuously retained in the multivariate analysis, suggesting that they could serve as separate prognostic indicators after the adjustments for other interference factors (Figure 9B). To discern the advantage of risk signature in terms of predicting OS over other clinical features including gender, age, white blood cell (WBC) count, blast cells percentage in bone marrow (BM), and cytogenetic risk, we integrated ROC curves for different clinical features in the Figure 9C. The AUCs for 5-year OS showed that the sensitivity and specificity of the risk signature was the best of all. Next, we tried to apply the risk signature to specific clinical scenarios by generating an intuitive visual nomograph based on the five clinical features along with the risk score using the ‘rms’ package (Figure 9D). The AUCs for 1-year, 3-year, and 5-year OS were 0.806, 0.842, and 0.909, respectively, indicating an outstanding discrimination accuracy of the merged score (Figure 9E). A calibration plot was further drawn to demonstrate a high degree of agreement between the predicted and observed 1-year, 3-year, and 5-year OS (Figure 9F).

## 4. Discussion

Expectancy of AML treatment is remarkably influenced by a wide range of intrinsic genomic changes and molecular mutations. The eradication of microresidual disease (MRD) after the achievement of remission to prevent relapse remains a tricky problem. Although allogeneic hematopoietic stem cell transplantation (alloHSCT) is considered to have the ability of eliminating MRD through the graft-versus-host disease, evidence shows that leukemic cells could evade the specific allogeneic immune response following transplant [21]. Meanwhile, a growing number of studies have revealed that the immune microenvironment in the bone marrow of AML patients underwent great changes compared to the healthy individuals [22]. In light of the complex molecular interactions between the tumor immunology and oncogenesis or resistance of AML, immunotherapy has been an object of intensive investigation. In addition, combined with clinicopathological and genetic features, immune factors could function as potent predictors of OS or event-free survival (EFS).

Regarded as a fail-safe mechanism when apoptosis is inhibited, necroptosis is featured by the release of cell materials and a large number of pro-inflammatory factors after the rupture of membrane. The inflammatory microenvironment formed by the considerable inflammatory cytokines released was found to have a dual effect on both the anti-tumor response and leukemogenesis [23]. For one thing, stimulated normal cells could be converted into tumor cells, which took control in the early stage of leukemogenesis [24]. For another, as reported in a recent study, the triggered inflammatory response could improve the tumor microenvironment hence igniting the specific anti-leukemia immune response [25]. In addition to the formation of an inflammatory microenvironment, mounting lines of evidence have suggested a direct interaction between necroptosis and immune cells. A study conducted by Kang et al. [26] showed impaired NKT cell activity in the *RIPK3* deletion mice, drawing the conclusion that *RIPK3* was involved in the regulation and promotion of NKT cell-mediated anti-tumor immunity. The development of therapeutic target of necroptosis could be promising given that it could be harnessed in the induction of tumor cell death and the boost of anti-tumor immunity. Nevertheless, effectiveness of immunotherapy in the limited subgroups and the off-target toxicity were still challenging the development of immunotherapy in AML. In this study, we generated a molecular signature based on the necroptosis-related genes by combining the clinical characteristics with the transcriptome data to explore whether the prognostic of AML patients was associated with necroptosis-related genes and to identify the particular subgroups sensitive to the immunotherapy as a novel biomarker.

To comprehensively delineate the complex role of necroptosis in a variety of tumors, a total of 111 genes were collected as necroptosis-related genes and the expression of these genes were compared between the AML and normal samples. Univariate and multivariate Cox were subsequently performed on the differentially expressed genes and the prognostic signature was established in the training set consisting of 6 genes.

*ZBP1* is considered a key component of the innate immune system as it recognizes and binds to exogenous Z-DNA or Z-RNA to trigger the downstream immune response. It is also another critical *RIPK1*-independent modality to drive necroptosis with the exception of *TNF*-induced necroptosis. In the present study, *ZBP1* was related to adverse prognostic with a high expression in the high-risk group, suggesting that the level of inflammatory response in the high-risk group was much higher.

*BRAF* is known as an oncogene as mutation in this gene drives tumorigenesis in a lot of cancer types including colorectal cancer, thyroid carcinoma, melanoma, etc. Recently, a researcher investigated the mechanism of the low expression of *RIPK3* in most cancer cell lines and found that *BRAF* along with *AXL* were two main oncogenes responsible for the loss of *RIPK3* repression and resistance of cancer cells to necroptosis [27]. Here, *BRAF* was linked with extended survival time in the low-risk group for its relative high expression and it was significantly downregulated in AML samples, allowing it to be a protective gene in the study.

*ALK* is a member of insulin receptor superfamily and exerts promoting effects on specific neurons in the nervous system. Coinciding with *BRAF*, *ALK* has been considered as a driving event in several cancer types due to its mutation and amplification, while the expression of *ALK* in AML samples and high-risk group is contrarily elevated, making it a risk factor to indicate oncogenesis and poor prognostic.

*BNIP3* plays a significant role in the induction of apoptosis pathway and is typically silenced in the tumors to develop apoptosis resistance. Song et al. [28] demonstrated that *BNIP3* was upregulated in the *RIPK3*-mediated cardiomyocytes necroptosis achieved by HR injury, making it an inducer in both apoptosis and necroptosis. We found that *BNIP3* was upregulated and implicated favorable prognostic in the normal samples and low-risk group, which might be a result of immune responses caused by *BNIP3*-mediated necroptosis.

*CDC37* functions as a molecular chaperone and is involved in the cell signal transduction and complex formation with *HSP90*. Li et al. [29] found in their study that *CDC37* together with *HSP90* constitute the complex which physically interacts with *RIPK3* and that necroptosis is inhibited in the *CDC37* deletion cells. In this study, the high expression of *CDC37* implied its contribution to the comparatively high activation of necroptosis in the high-risk group.

*YBX3* belongs to the Y-box binding protein family and facilitates the RNA-binding activity. It plays a regulatory role in a wide spectrum of biological processes including negative regulation of programmed cell death. Here, *YBX3* functioned as a tumor suppressor gene as it was a protective factor in the regression model and expressed higher in the normal samples at meantime.

To conclude, *BRAF*, *ALK*, and *YBX3* can be classified as inhibitor of necroptosis whereas the left three genes promoters. Notably, two promoters along with one inhibitor are upregulated in the high-risk group and two inhibitors along with one promotor are upregulated in the low-risk group. The frequently conflicting results might be attributed to the interactions between the seven genes and further investigations are required.

The two risk subgroups generated on the basis of median risk score distinguish significantly in terms of clinical prognostic, signaling pathways, immune microenvironment, and response to immunotherapy. Pathway enrichment analysis results demonstrated that the uniquely upregulated genes in the high-risk group are mainly involved in various immune-related pathways usually activated in innate or adaptive immunity, suggesting that the level of immune response in the high-risk group is relatively higher. However, patients in the high-risk group do not enjoy a prolonged survival time which was normally associated with the enhanced immune response. Reasons accounting for the contradictory fact may lie in that necroptosis is known to play both pro-tumor and anti-tumor roles. The two effects should coexist during the necroptosis process and which effect dominates depends on the development stage of the tumor. We speculate that the necroptosis-induced inflammation advances the progression of tumor in the early stage with the inflammatory cytokines or growth factors which enhance the proliferation and resistance of tumor cells. Then, with the formation of inflammatory microenvironment, the anti-tumor immune response is stimulated and takes control through the recruitment of immune cells to the TME or promotion of the antigen-presentation by mature dendritic cells, macrophages, etc. [26]. The immune activation-related cells are more infiltrated in the high-risk group, which also confirmed the presence of a more active anti-tumor immunity. In this case, we assume that in spite of the disadvantage over the OS, there may be an advantage over the immunotherapy response in the high-risk group. Immunotherapy by immune checkpoint blockade has arisen as a new milestone for a number of tumors of which therapeutic options were previously limited. The expression of several classical checkpoint genes was explored in this study, and it turned out that all genes with significant difference were overexpressed in the high-risk group including the intensively studied *PD-1* and *CTLA4*. The original TIDE algorithm provides a more accurate method than the single checkpoint gene expression level to predict patients’ response to ICB. Consistent with the checkpoint genes level, the TIDE results classify patients with high-risk score as responders to immunotherapy, which supports our speculation to some extent. However, in-depth studies are still demanded to seek robust proof and to investigate the underlying mechanisms of necroptosis-induced immunity in AML.

## 5. Conclusions

In summary, our study focuses on the generation of the risk signature based on the necroptosis-related genes and the exploration of its prognostic and immunological value. First of all, we comprehensively analyzed the expression of necroptosis-related genes in AML patients. The signature has been proven to be an independent risk factor to predict prognostic in AML and its predictive performance is superior to the normal clinicopathological features. Moreover, we directly link necroptosis with clinical prognostic and promoters of necroptosis identified in this study can provide some insights for the targeted therapy. Finally, we elucidate the immune features and immunotherapy response in the different subgroups which is instrumental to the patient selection and clinical management for the optimization of immunotherapy. A rational combination of immunotherapy such as ICB with chemotherapeutics and necroptosis-targeted therapy seems a promising strategy which deserves further study. Yet, there are certain limitations in the study. First, our study is a retrospective analysis centered on the public databases, such that in vivo and in vitro experiments are urgently needed to validate the practical value of the signature. Additionally, the finite sample size of the datasets may limit the extensiveness of the signature in other AML populations. Moreover, the corresponding AML immunotherapy data should be gathered and excavated in the future to validate potential of the signature in informing immune response.

## Figures and Tables

**Figure 1 genes-13-01837-f001:**
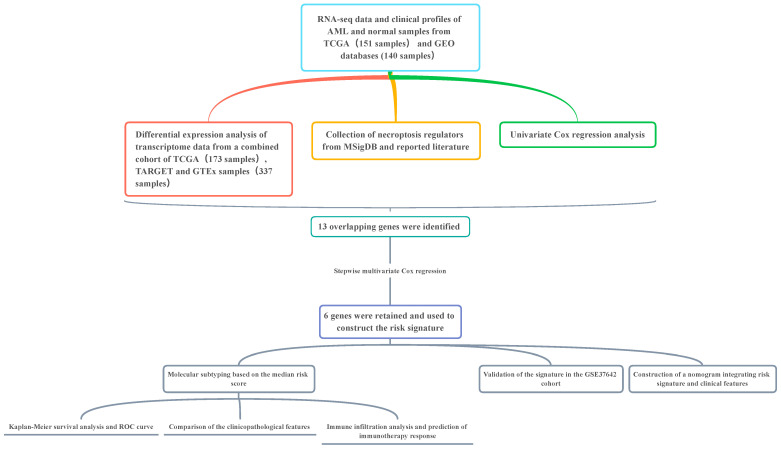
Overall flow chart of research design.

**Figure 2 genes-13-01837-f002:**
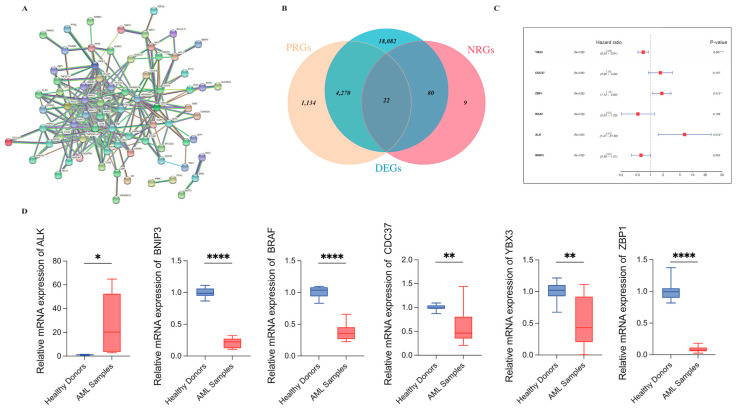
Filtering of differentially expressed necroptosis-related genes of prognostic value. (**A**) Molecular interactions of 111 necroptosis-related regulators depicted by the STRING platform. (**B**) Venn plot shows integration of 3 groups of genes and the overlapping part consists of 22 genes. (**C**) 6 genes were eventually screened through the multivariate Cox regression to generate the prognostic model. *p* values are showed as: * *p* < 0.05; ** *p* < 0.01; **** *p* < 0.0005. (**D**) The real-time PCR results demonstrated the mRNA levels of 6 genes in the risk model in healthy donors (n = 3) and AML samples (n = 4). (Student *t*-test, * *p* < 0.05; ** *p* < 0.01; **** *p* < 0.0005).

**Figure 3 genes-13-01837-f003:**
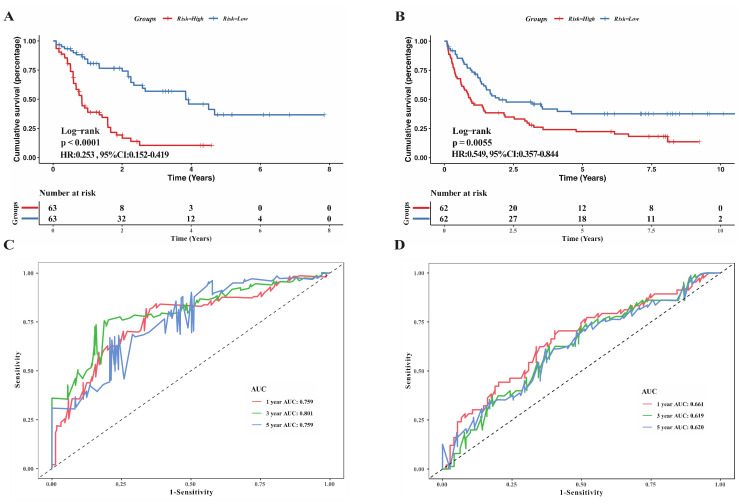
Assessment of ability of the risk signature to predict prognostic in both training and validation cohorts. (**A**,**B**) Kaplan-Meier survival curves reveal the difference of prognostic in two risk subgroups in TCGA cohort (**A**) and GSE37642 cohort (**B**). (**C**,**D**) ROC curves showing the predictive power of the risk model on the survival rate in TCGA cohort (**C**) and GSE37642 cohort (**D**).

**Figure 4 genes-13-01837-f004:**
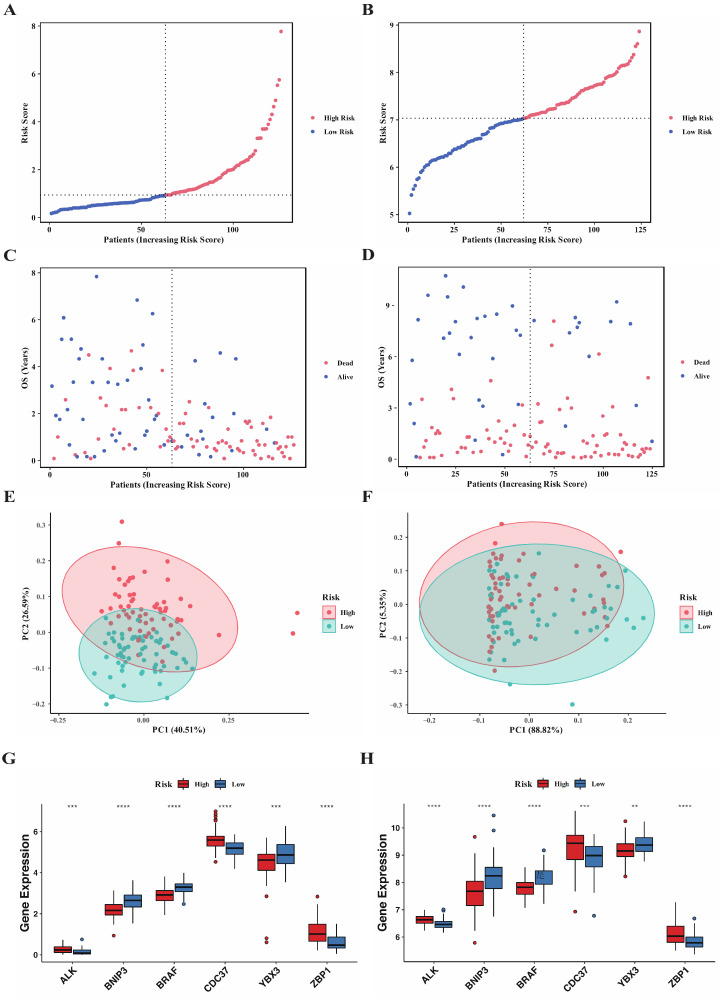
Two risk subgroups generated based on the median risk score. (**A**–**D**) Distribution of risk score, survival time and survival status for AML patients in TCGA cohort (**A**,**C**) and GSE37642 cohort (**B**,**D**). (**E**,**F**) Principal component analysis of the risk score to distinguish the two subgroups in TCGA cohort (**E**) and GSE37642 cohort (**F**). (**G**,**H**) Comparison of the expression of 6 genes in the risk model between the two subgroups in TCGA cohort (**G**) and GSE37642 cohort (**H**). (Wilcox test, ** *p* < 0.01; *** *p* < 0.001, **** *p* < 0.0005).

**Figure 5 genes-13-01837-f005:**
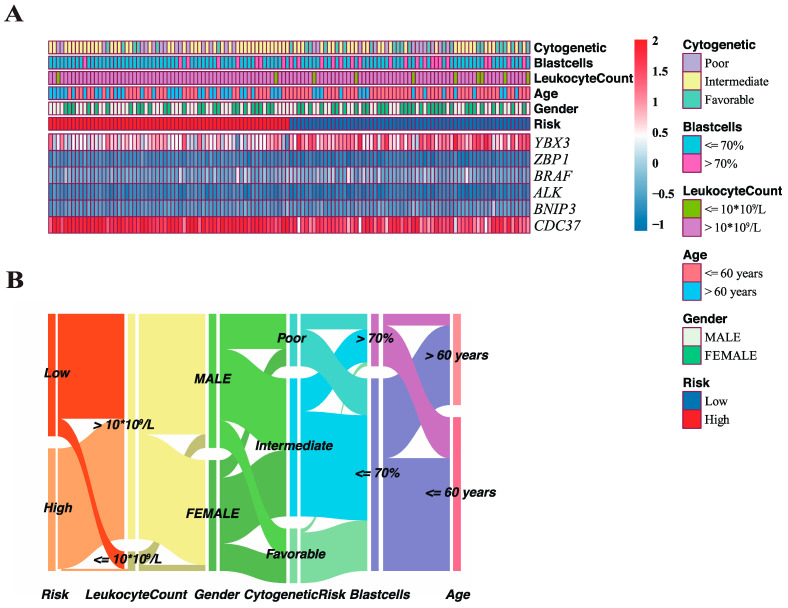
Characteristics of classical clinical features in two risk subgroups. (**A**) Heatmap visualizing the expression of 6 genes and clinicopathological features at different risk levels in TCGA cohort. (**B**) Alluvial diagram demonstrating a dynamic network of the interrelated clinical features in TCGA cohort.

**Figure 6 genes-13-01837-f006:**
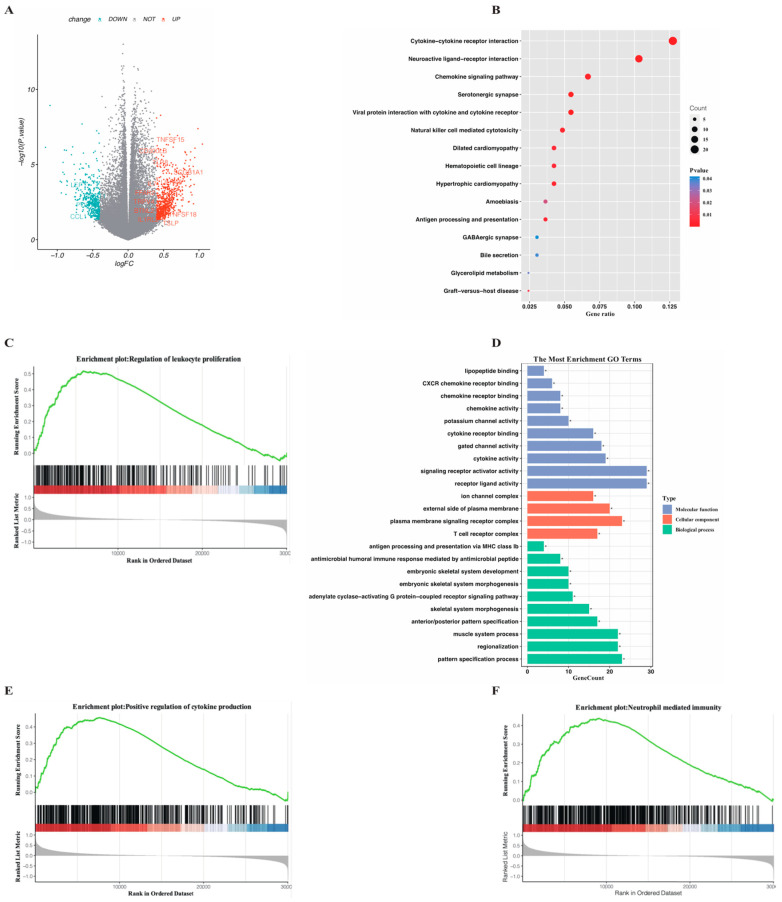
Functional enrichment analysis performed on the DEGs identified between the two risk subgroups in TCGA cohort. (**A**) Volcano plot displaying DEGs between the two risk subgroups by the threshold of |log2FC| ≥ 0.4 and adjusted *p*-value < 0.01. Specific DEGs involved in the selected pathways are displayed. (**B**) KEGG pathways significantly enriched in the high-risk group in the bubble plot form. (**C**,**E**,**F**) Selected canonical biological process associated GSEA pathways in the high-risk group. (**D**) Representative GO terms enriched in terms of biological process (BP), cellular component (CC) and molecular function (MF) respectively in the high-risk group. *p* value is showed as: * *p* < 0.05.

**Figure 7 genes-13-01837-f007:**
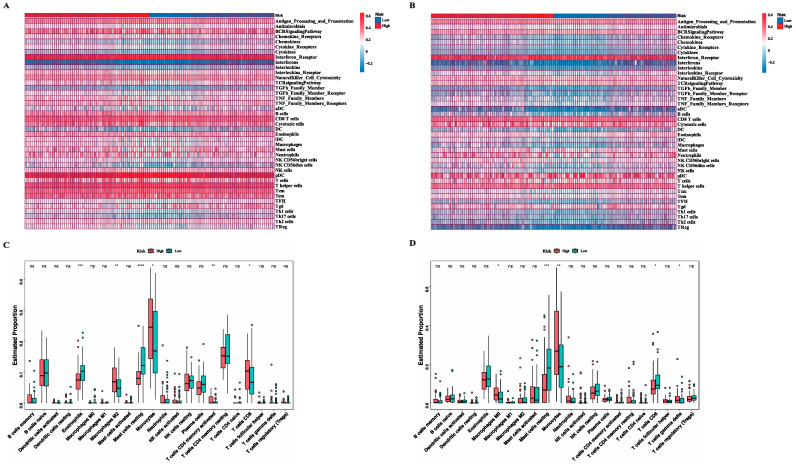
Landscape of immune activity and relevant infiltrating immune cells in the two risk subgroups. (**A**,**B**) Heatmap showing the ssGSEA enrichment scores of 40 immune components including immune cells, relative immune factors and pathways in TCGA (**A**) and GSE37642 (**B**) cohorts. (**C**,**D**) Comparisons of proportion of 22 immune cells between the two risk subgroups in TCGA (**C**) and GSE37642 (**D**) cohorts (Wilcox text, * *p* < 0.05; ** *p* < 0.01; *** *p* < 0.001, **** *p* < 0.0005).

**Figure 8 genes-13-01837-f008:**
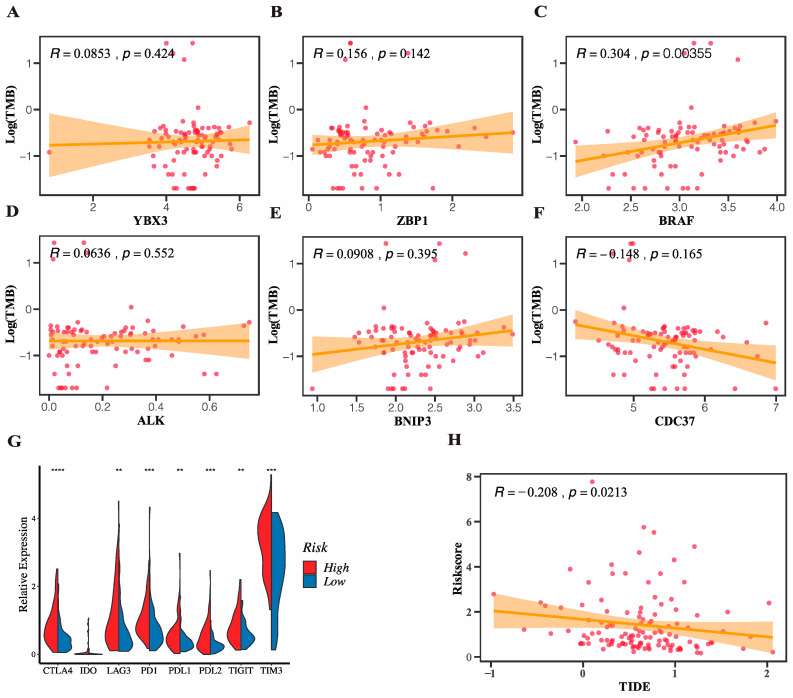
Evaluation of potential value of the risk model in the ICB immunotherapy in TCGA cohort. (**A**–**F**) Correlation of the 6 genes in the signature with TMB. (**G**) Differences of expression profiles of common checkpoint genes in the two risk subgroups (Wilcox text, ** *p* < 0.01; *** *p* < 0.001, **** *p* < 0.0005). (**H**) Correlation between the risk score and TIDE score (Spearman test, *p* < 0.05).

**Figure 9 genes-13-01837-f009:**
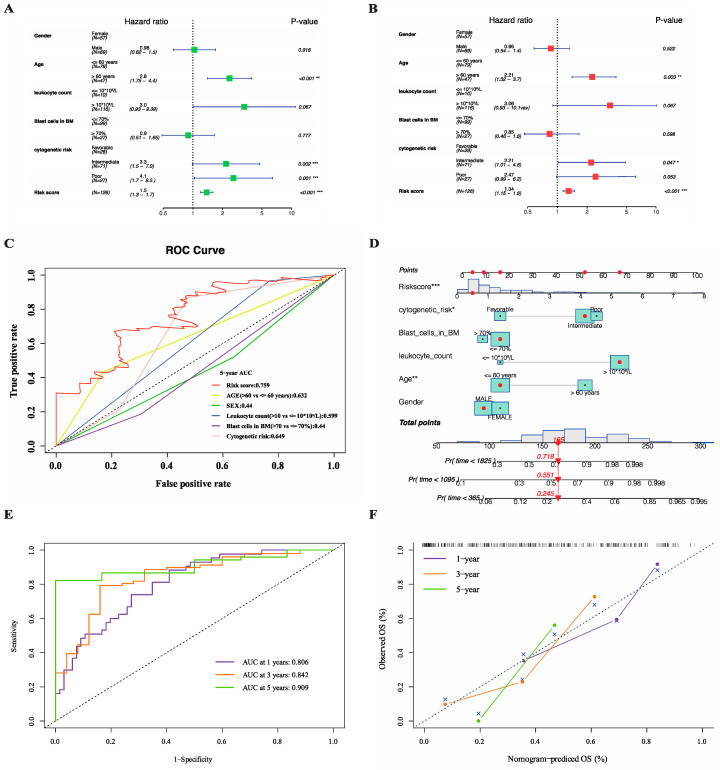
Construction of the OS-predictive nomogram for clinical application in TCGA cohort. (**A**,**B**) Univariate (**A**) and multivariate (**B**) Cox regression analysis of the risk signature to verify its independent predictive efficacy of OS. (**C**) ROC curves for the risk score and major clinical features including gender, age, white blood cell (WBC) count, blast cells percentage in bone marrow (BM) and cytogenetic risk. (**D**) Nomogram based on the combination of clinical features and risk score predicting 1-year, 3-year and 5-year survival. (**E**) ROC curves showing the predictive power of nomogram. (**F**) Calibration plot showing the predictive reliability of nomogram. *p* values are showed as: * *p* < 0.05; ** *p* < 0.01; *** *p* < 0.001.

## Data Availability

TCGA-TARGET-GTEx cohort: https://xenabrowser.net/datapages/?cohort=TCGA%20TARGET%20GTEx&removeHub=https%3A%2F%2Fxena.treehouse.gi.ucsc.edu%3A443 GDC TCGA Acute Myeloid Leukemia (LAML) cohort: https://xenabrowser.net/datapages/?cohort=GDC%20TCGA%20Acute%20Myeloid%20Leukemia%20(LAML)&removeHub=https%3A%2F%2Fxena.treehouse.gi.ucsc.edu%3A443 GSE37642 cohort: https://www.ncbi.nlm.nih.gov/geo/query/acc.cgi?acc=GSE37642 GSE6891 cohort: https://www.ncbi.nlm.nih.gov/geo/query/acc.cgi?acc=GSE6891 GSE111678 cohort: https://www.ncbi.nlm.nih.gov/geo/query/acc.cgi?acc=GSE111678 GSE71014 cohort: https://www.ncbi.nlm.nih.gov/geo/query/acc.cgi?acc=GSE71014 GSE114868 cohort: https://www.ncbi.nlm.nih.gov/geo/query/acc.cgi?acc=GSE114868.

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
