# Peer review of "A Novel Identified Necroptosis-Related Risk Signature for Prognosis Prediction and Immune Infiltration Indication in Acute Myeloid Leukemia Patients"

_genes, 2022, doi:10.3390/genes13101837_

Round 1

Reviewer 1 Report

Dear Authors, 

please find comments on the manuscript below:

1. Abstract: Please indicate which genes were selected as s risk signature. In addition, specific values like HR, ROC score, etc. should be provided in the Abstract.

2. Page 3, lines 126, 127: 

The sentence "Method selected for the stepwise regression for variables screening was stepwise" should be explained. What kind of stepwise method was chosen? Backward? Forward? 

3. Page 3, line 140: Please, explain why the cut-off value was set up as >>0.4

4. Page 3, line 147: What kind of public resources did you use? Please indicate it. In the Results section (page 5, line 19), you mentioned pubic literature and the ImmPort database. 

Reviewer 2 Report

In the article “A Novel Identified Necroptosis-Related Risk Signature for Prognosis Prediction and Immune Infiltration Indication in Acute Myeloid Leukemia Patients” Sun et al., used Bioinformatics tools and identified a necroptosis-related signature as a novel biomarker to improve the risk stratification, to inform the immunotherapy efficacy and to indicate the therapeutic option of targeted therapy. The overall study was impressive, however, the below comments need to be addressed.

Major comments:

1.      In Figures 4G and 4H, the YBX3 gene expression was higher in a low-risk group compared to the low-risk group, but in Figure 5A in the heat map, they showed the YBX3 gene expression was higher in the high-risk group? The authors need to clarify this.

2.      Even though CDC37 (is a specific co-chaperone and oncogenic kinase recruiter for a diverse group of protein kinases) was shown higher in a high-risk group, the CDC37 expression should be shown at the protein level, in AML patients’ samples or AML cell lines.

3. Similarly, the PD-1/PD-L1, CTLA4, and TIM3 immune checkpoint levels should be shown in ANM cell lines or in AML patients' samples to correlate with the bioinformatics analysis presented in this study. 

Round 2

Reviewer 2 Report

Even though the authors did not perform the suggested in vitro experiments, they find some alternative methods from publically available databases to validate the expression of CD37, PD/PDL1, TIM3, and CTLA4 in two risk groups and showed consistent results with their findings. Overall the revised manuscripts look impressive and included all other suggested concerns, therefore I recommend accepting this manuscript for possible publication in Genes.